# DispViT: Direct Stereo Disparity Regression with a Single-Stream Vision Transformer

**Tongfan Guan**[1]**, Jiaxin Guo**[1]**, Tianyu Huang**[1]**, Jinhu Dong**[1]**, Chen Wang**[2]**, Yun-Hui Liu**[1,†]

[1]The Chinese University of Hong Kong    [2]University at Buffalo    [†]Corresponding Author

## Abstract

Deep stereo disparity estimation has long been dominated by a **matching-centric paradigm**, which constructs cost volumes and iteratively refines local correspondences. Despite its success, this formulation exhibits an intrinsic vulnerability: visual ambiguities from occlusion or non-Lambertian surfaces inevitably induce erroneous matches that local refinement cannot fully recover. This paper introduces **DispViT**, a new architecture that establishes a **regression-centric paradigm**. Instead of explicit matching, DispViT directly regresses disparity from tokenized binocular representations using a single-stream Vision Transformer. This is enabled by a set of lightweight yet critical designs, such as a probability-based disparity parameterization for stable training and an asymmetrically initialized stereo tokenizer for effective view distinction. To better align the two views during stereo tokenization, we introduce a novel shift-embedding mechanism that encodes multiple disparity shifts into channel groups, preserving geometric cues even under large view displacements. A lightweight refinement module then sharpens the regressed disparity map for fine-grained accuracy. By prioritizing holistic regression over explicit matching, DispViT streamlines the stereo pipeline while improving robustness and efficiency. Extensive experiments on standard benchmarks demonstrate state-of-the-art accuracy, along with strong resilience to matching ambiguities and wide disparity ranges.

## 1 Introduction

Stereo disparity estimation is one of the core challenges in computer vision, with applications in autonomous driving (Geiger et al., 2013), augmented reality (Kim et al., 2018), and robotic manipulation (Fang et al., 2023). The objective is to compute the horizontal displacement of pixels between two rectified images from a stereo camera rig. A dominant paradigm for stereo disparity estimation has been matching-centric, which explicitly establishes pixel-level correspondence between the left and right views. This perspective has driven the prevalence of pipelines (Žbontar & LeCun, 2016; Kendall et al., 2017; Lipson et al., 2021; Li et al., 2021) built around cost volumes and iterative refinement. While effective, the *matching-centric* paradigm exhibits an intrinsic limitation: matching is inherently ill-posed in the presence of visual ambiguities such as transparency, occlusion, or repeated patterns. Moreover, unreliable matches are often difficult to recover via subsequent local refinements, leaving the pipeline brittle in cases where robustness is critical. This motivates us to rethink stereo disparity estimation from a different perspective—one that bypasses explicit matching.

Currently, Vision Transformers (ViTs) have demonstrated remarkable capabilities in geometry regression tasks like monocular depth estimation (Yang et al., 2024; Piccinelli et al., 2024) and feedforward 3D reconstruction (Wang et al., 2024a; 2025). However, the use of ViTs in stereo networks has been largely confined to view feature extractors (Wen et al., 2025; Liu et al., 2024) within conventional matching pipelines, leaving their potential for direct stereo disparity regression largely unexplored. In this work, we advocate a *regression-centric* perspective: rather than building increasingly elaborate cost volumes and refinement mechanisms, we harness the global reasoning capacity of a ViT (Dosovitskiy et al., 2020) to perform direct disparity regression through holistic analysis of visual context and binocular cues. This holistic regression yields a strong initial estimate, which we complement with a lightweight refinement module for fine-grained accuracy. By rethinking stereo

---

Code available at: `https://github.com/aeolusguan/DispViT`.

disparity estimation around regression rather than matching, our approach circumvents the core vulnerability of matching pipelines: their susceptibility to visual ambiguities.

This paradigm shift raises a key design question: how should binocular images be tokenized for a ViT to enable direct regression? Conventional matching-centric approaches encode the two views separately (Li et al., 2021; Weinzaepfel et al., 2023) for establishing matching. Departing from this dual-stream manner, our regression-first philosophy pursues a single-stream formulation. The pioneering regression-centric work of DispNetS (Mayer et al., 2016) simply concatenated stereo pairs along the channel dimension for a CNN to regress disparity. While conceptually elegant, this method was fundamentally limited by the localized receptive field of convolution, which hindered effective reasoning over large disparities or complex global contexts, ultimately constraining generalization. The global attention mechanism of ViTs offers an effective remedy, enabling holistic reasoning across both views without the locality bottleneck. However, directly concatenating the two views poses its own challenge: significant pre-attention misalignment at the token level caused by large disparities, which corrupts the ViT's input and impedes its ability to infer binocular geometry. This motivates a new stereo tokenization design that is expected to mitigate early misalignment and allow a single-stream ViT to operate effectively.

To this end, we propose a shift-embedding stereo tokenizer that mitigates input-level misalignment by horizontally shifting the right view with a set of predefined offsets. Each shifted variant is independently tokenized into a separate channel group and then blended with the left-view tokens, allowing each spatial token to encode a spectrum of potential alignments and easing reasoning over large disparities. Complementing this, we introduce a simple yet effective asymmetric initialization of the patchification convolution, which distinguishes the left and right views from the earliest stage of training and prevents early degeneracy. To further embed binocular geometry priors into holistic reasoning, we extend Rotary Position Embeddings (RoPE) (Su et al., 2024) to a disparity-aware formulation (DA-RoPE), enabling aggregated features to remain geometry-consistent even at large displacements. Together, these lightweight yet critical components form the foundation of DispViT, a single-stream Vision Transformer that successfully bypasses explicit matching and directly regresses disparity from a holistically reasoned binocular representation, as depicted in Figure 1.

Our DispViT, pretrained on a large corpus of data, demonstrates strong robustness to matching ambiguities (see Figure 4). When complemented with a lightweight refinement module for fine-grained details, it achieves state-of-the-art performance while maintaining efficiency. In summary, we introduce DispViT, the first single-stream ViT framework that bypasses explicit matching and directly regresses stereo disparity from tokenized binocular representations. At its core is a single-stream ViT backbone, equipped with shift-embedding stereo tokenizer, probability-based parameterization of disparity, asymmetric initialization, and Disparity-Aware RoPE (DA-RoPE). This regression-first formulation provides a strong disparity initialization that, together with lightweight refinement, establishes a robust and efficient alternative to long-standing matching-centric pipelines.

## 2 RELATED WORK

**Deep Stereo Matching.** The foundation of modern deep stereo matching is built upon the pipeline that first extracts discriminative features from a binocular pair and then establishes matching through cost volume or cross-view attention. Seminal works such as GC-Net (Kendall et al., 2017) and PSM-Net (Chang & Chen, 2018) pioneered the 3D cost volume architecture: using 2D CNNs for feature extraction, constructing a 3D volume, and processing it with 3D convolutions for cost aggregation. Subsequent research has extensively refined this framework by incorporating richer contextual information (Xu et al., 2022; Shen et al., 2022; 2021) and developing more powerful aggregation networks (Zhang et al., 2019; Guan et al., 2024). RAFT-Stereo (Lipson et al., 2021) fundamentally shifted the matching paradigm by replacing explicit cost volume processing with a recurrent decoder that queries a pre-computed, multi-scale 4D correlation space for iterative disparity refinement. This paradigm evolved through innovations like IGEV's geometric encoding (Xu et al., 2023a), the frequency decomposition of DNLR (Zhao et al., 2023) and Selective-Stereo (Wang et al., 2024b), and Mocha-Stereo's motif-based attention (Chen et al., 2024), collectively improving accuracy, robustness and generalization. In contrast with iterative local refinement, another line of research (Li et al., 2021; Xu et al., 2023b; Weinzaepfel et al., 2023; Min et al., 2025) employs cross-view attention between separately encoded view features to perform global matching.

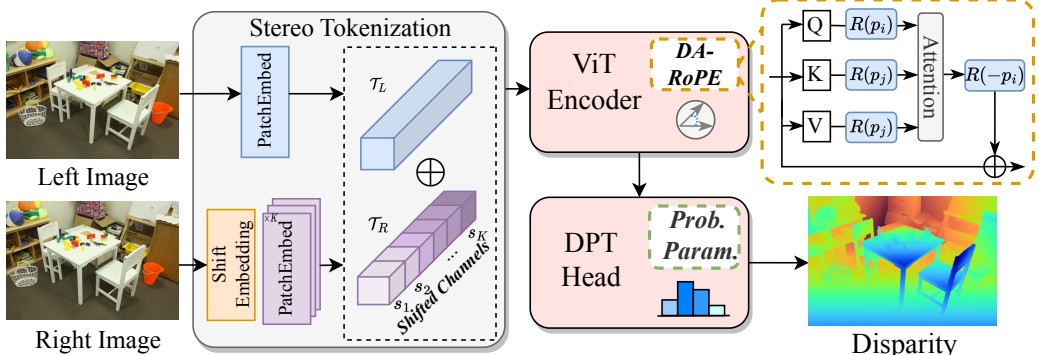

Figure 1: **Overview of DispViT.** We introduce a regression-centric paradigm for stereo disparity estimation using a simple single-stream ViT. The effectiveness of this simple architecture is enabled by lightweight yet critical designs, such as probability-based parameterization of disparity and stereo tokenizer exemplified here, among other critical components explored in the text.

Recognizing the inherent ambiguities of matching, a growing line of research leverages monocular depth estimation to bootstrap stereo. Recent approaches such as Monster (Cheng et al., 2025) and DEFOM-Stereo (Jiang et al., 2025) initialize RAFT-style iterative refinement with a scale-aligned monocular disparity map. This initialization, inherently free from matching ambiguities, provides a strong prior that markedly enhances both the accuracy and robustness of stereo disparity estimation. The concurrent BridgeDepth (Guan et al., 2025) unifies monocular and stereo reasoning through iterative bidirectional alignment of latent representations, efficiently synthesizing stereo precision with monocular robustness. All these advances harness the core advantage of monocular disparity regression: immune to matching ambiguities. The success of these hybrid approaches inspires our pivotal departure—to directly regress disparity from the binocular input without explicit matching.

**ViTs for Correspondence.** ViTs have demonstrated strong capacity in modeling visual correspondence, typically following an encoder-aggregator architecture. Feed-forward 3D reconstruction methods such as MASt3R (Leroy et al., 2024) and VGGT (Wang et al., 2025) ground image matching in 3D reconstruction, producing structure-aware dense features to establish cross-view correspondences. Furthermore, CrocoV2 (Weinzaepfel et al., 2023) and UFM (Zhang et al., 2025) directly predict correspondence fields with a DPT (Ranftl et al., 2021) head after feature aggregation using cross-attention or alternate-attention. In stereo matching, FoundationStereo (Wen et al., 2025) employs a ViT as the feature extractor, complemented by a CNN to fuse global context with fine details. Despite architectural variations, these approaches are inherently matching-centric, focusing on aligning features across images. In contrast, DispViT departs from the encoder-aggregator design and introduces a single-stream ViT that directly regresses disparity from binocular input, recasting stereo matching as regression rather than correspondence search.

## 3 METHOD

Given a rectified stereo pair $(\mathcal{I}_L, \mathcal{I}_R) \in \mathbb{R}^{H \times W \times 3}$, our goal is to predict the disparity map of the left view $\mathcal{D} \in \mathbb{R}^{H \times W}$. To this end, we propose a transformer-based architecture that directly regresses the disparity map from a unified token representation of the two views using a single-stream ViT:

$$\hat{\mathcal{D}}_0 = \text{DPT}\big(\Phi \circ \mathcal{T}(\mathcal{I}_L, \mathcal{I}_R)\big), \tag{1}$$

where $\mathcal{T}$ tokenizes the binocular input into a single sequence and $\Phi$ denotes a ViT enhanced with a novel disparity-aware Rotary Positional Embeddings (RoPE). A DPT (Ranftl et al., 2021) head fuses the transformer's multi-scale features to regress an initial disparity map $\hat{\mathcal{D}}_0$, which is subsequently sharpened by a lightweight refinement module to produce the final prediction $\hat{\mathcal{D}}$.

### 3.1 STEREO TOKENIZATION

Stereo tokenization is the crucial first step for enabling direct disparity regression within a single-stream ViT. It patchifies the left and right views and blends them into a single sequence. To inherit

the power of pretrained models, the `PatchEmbed` module of DINOv2 (Oquab et al., 2023) is adapted to handle binocular input. This standard tokenization layer, denoted as $\mathcal{E}$, is essentially a strided 2D convolution, mapping 3-channel RGB to patchified high-dimensional embeddings. A straightforward extension would be to concatenate the two views along the channel dimension, duplicate the convolution weights, and scale them by half, as in Marigold (Ke et al., 2024) when adapting Stable Diffusion to monocular depth estimation.

However, we found **asymmetric initialization** yields substantially superior performance in practice. Concretely, we initialize the new convolution kernel by concatenating the original pretrained weights with a zero tensor of identical shape instead of duplicating and halving them. We hypothesize this "half-zero" initialization provides a critical inductive bias: the pretrained branch processes the left view as a clear, stable reference, while the zero-initialized branch is compelled to learn specialized features that complement the reference from the right view. This asymmetry encourages the model to distinguish the two views from the first layer, a property that symmetric initialization lacks.

Furthermore, large disparities introduce spatial misalignment between the two views. Direct channel concatenation in this case mixes features from unrelated image regions, leading to incoherent token representations. To mitigate this, we design a **shift-embedding** tokenizer that encodes multiple alignment hypotheses within each token. Let $\{s_k\}_{k=1}^{K}$ be a set of predefined horizontal shift offsets. For each offset $s_k$, the right image $\mathcal{I}_R$ is shifted by $s_k$ and passed through a specific convolution $\mathcal{E}_R^k$, producing channel groups

$$\mathcal{T}_R^{(k)} = \mathcal{E}_R^k\big(\mathtt{Shift}(\mathcal{I}_R, s_k)\big), \quad k = 1, \cdots, K. \tag{2}$$

These groups are concatenated along the channel dimension to form the right-view embedding $\mathcal{T}_R = \mathtt{Concat}\big(\{\mathcal{T}_R^{(k)}\}_{k=1}^{K}\big)$. Meanwhile, the left view $\mathcal{I}_L$ is tokenized by the pretrained `PatchEmbed` $\mathcal{E}$. The asymmetric initialization is still employed, *i.e.*, $\mathcal{E}$ retains the pretrained weights while $\{\mathcal{E}_R^k\}_{k=1}^{K}$ are initialized with zeros. Finally, the stereo tokens are blended pixelwise by summation, $\mathcal{T} = \mathcal{T}_L + \mathcal{T}_R$, yielding a unified token sequence in which each spatial token embeds a spectrum of potential disparities. This design preserves disparity structure at the input level and facilitates ViT's reasoning over large displacements. Compared to direct channel concatenation, our shift-embedding tokenizer incurs negligible overhead since $\mathcal{T}_R$ can be implemented using an optimized groupwise convolution, while outperforming with an appreciated margin.

Although shift-embedding introduces multiple horizontally shifted variants of the right view, it neither constructs nor approximates a cost volume. Unlike cost volumes, which explicitly compute pairwise feature similarities across disparity hypotheses, shift-embedding performs no correlation or cost aggregation, but instead encodes coarse alignment cues into channel groups during tokenization. Disparity is then directly regressed from tokenized stereo input by a single-stream ViT, making shift-embedding an alignment prior rather than an explicit matching mechanism.

## 3.2 SINGLE-STREAM VISION TRANSFORMER

At the core of our architecture lies a single-stream Vision Transformer (ViT), which unifies feature extraction and correspondence reasoning within a single Transformer backbone, thereby bypassing the need for explicit matching modules. We build upon a pretrained DINOv2 ViT backbone, capitalizing on its robust visual representations while ensuring compatibility with our stereo tokenization scheme. The transformed multi-scale features are then consumed by a DPT head to predict disparity.

To provide the ViT with spatial awareness, DINOv2 adds learnable absolute positional embeddings (APE) to each token. However, we found APE ill-suited for disparity regression. We hypothesize the issue lies in its lack of translational equivariance: disparity is inherently a relative offset, yet APE encodes only absolute locations without an effective mechanism to capture relative displacements. Unlike APE, Rotary Positional Embeddings (RoPE) (Su et al., 2024) encode positions by rotating queries and keys such that attention depends on relative offsets rather than absolute coordinates. This inductive bias aligns naturally with stereo geometry, where disparity manifests as horizontal translation. Empirically, we observe that substituting APE with RoPE brings substantial performance gains, underscoring the necessity of modeling relative geometry in disparity regression.

While RoPE ensures attention weights are translationally equivariant, it leaves the **value** vectors agnostic to relative position. But, for disparity estimation, the semantic meaning of a feature is

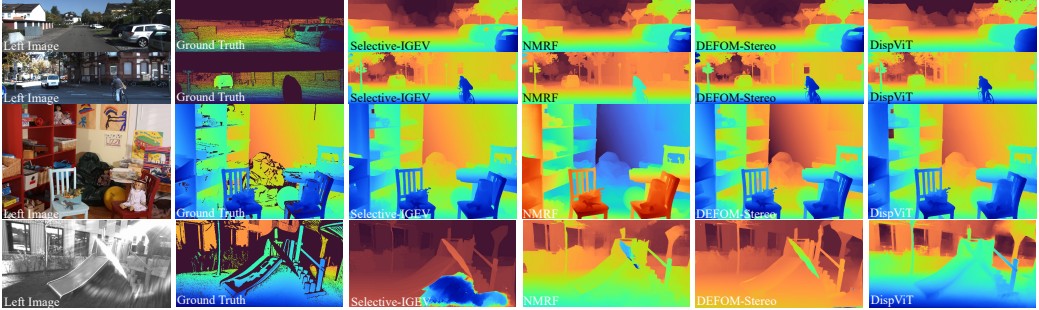

Figure 2: **Zero-shot generalization on real-world datasets.** Qualitative comparison with Selective-IGEV, NMRF, and DEFOM-Stereo across four datasets. Our DispViT (last column) exhibits superior robustness to matching ambiguities, including low-texture regions ("black holes") and complex surface materials like reflections and transparency. *Best viewed in color and zoomed in.*

intrinsically tied to its position relative to the viewer. Motivated by this intuition, we introduce a **Disparity-Aware RoPE** (**DA-RoPE**), which conditions value encoding on relative position. Concretely, each value $\mathbf{v}_j$ is rotated by its position $\boldsymbol{R}(\mathbf{p}_j)$, aggregated with the attention weights, and counter-rotated by the query position $\boldsymbol{R}(-\mathbf{p}_i)$. The resulting representation

$$\tilde{\mathbf{z}}_i = \boldsymbol{R}(-\mathbf{p}_i)\left(\sum_j \alpha_{ij}\boldsymbol{R}(\mathbf{p}_j)\mathbf{v}_j\right) = \sum_j \alpha_{ij}\boldsymbol{R}(\mathbf{p}_j - \mathbf{p}_i)\mathbf{v}_j \tag{3}$$

is equivalent to rotating each $\mathbf{v}_j$ by the relative position $\mathbf{p}_j - \mathbf{p}_i$ before aggregation. Intuitively, DA-RoPE re-expresses features in the query's local reference frame before aggregation, ensuring that both attention weights and aggregated features are consistently disparity-aware. This design embeds translational equivariance directly into the value pathway and equips the ViT with a stronger inductive bias for robust disparity estimation, especially under large disparities.

**Prediction head.** We adopt a DPT head to fuse multi-scale features from the ViT backbone for disparity estimation. A key design choice lies in the parameterization of disparity prediction. Instead of regressing disparity values directly, we discretize the disparity range into uniformly spaced bins and let the head output a probability distribution over these bins (inspired by Zholus et al. (2025)). The final disparity estimate is computed as the expectation over the distribution within a local window around the peak probability. This probabilistic formulation provides two advantages: it naturally reflects the bounded disparity range with a well-structured output space, and it allows the model to capture uncertainty in ambiguous regions rather than collapsing to a scalar value. Experiments suggest this **probability parameterization** is one of the most important components in our architecture. The network is supervised with a combined loss function that includes cross-entropy loss for the discrete distribution $\mathcal{P}$ and an L1 loss on the continuous estimate $\hat{\mathcal{D}}_0$ to ensure accuracy, *i.e.*,

$$\mathcal{L}_{\texttt{regress}} = \text{CE}\big(\mathcal{P}, \texttt{bilinear}(\mathcal{D}^*)\big) + \lambda_1 \text{L1}(\hat{\mathcal{D}}_0, \mathcal{D}^*), \tag{4}$$

where $\texttt{bilinear}(\mathcal{D}^*)$ denotes the bilinear assignment of the ground-truth disparity to discrete bins.

### 3.3 REFINEMENT

While the single-stream ViT generates a strong and robust disparity estimate, it inevitably misses fine-grained details, as no explicit two-view comparison is performed within the backbone. To recover these details, we introduce a lightweight refinement module applied after the direct regression. This module revisits the stereo pair and focuses on local correspondence cues, enabling sharper object boundaries and better reconstruction of thin structures. By design, it complements the ViT's global reasoning with precise local matching, yielding a complete and accurate disparity estimate.

To maintain efficiency, our refinement avoids reconstructing a cost volume. Instead, we adopt the refinement module of NMRF (Guan et al., 2024), which anchors local matching on a single-pass geometry warping guided by the initial disparity estimate, in contrast to the iterative cost-volume indexing of RAFT-style refinement. Specifically, the predicted disparity is used to warp the right image features toward the left view, producing aligned representations that highlight local inconsis-

tencies. A lightweight Swin Transformer (Liu et al., 2021) then integrates the warping feature fields with the initial disparity prediction to correct fine details.

**Decoupled training.** We adopt a two-stage training scheme to preserve modularity and flexibility. First, our single-stream ViT is trained independently using the loss presented in Equation 4, yielding a robust direct disparity regressor. Subsequently, we train the refinement model following the NMRF protocol, with the ViT regressor kept frozen. This decoupled strategy ensures that the ViT regressor can serve as a standalone model, directly deployable in applications where efficiency is paramount, or seamlessly integrated with external refinement modules to recover fine-grained details.

## 3.4 DISCUSSION

A central contribution of this work is establishing a strong baseline for direct disparity regression, which has traditionally been considered ill-posed due to the lack of explicit matching mechanisms. This capability is unlocked by several key designs. In particular: (1) The **probability parameterization** of disparity prediction significantly stabilizes training and boosts accuracy compared to scalar value regression; (2) Our **shift-embedding stereo tokenizer** preserves disparity structure in the blended token representation of two views; (3) The **Disparity-Aware RoPE** (**DA-RoPE**) extension equips ViT attention with the translational equivariance inductive bias essential for disparity matching; (4) **Asymmetric initialization** prevents early-stage training degeneracy and promotes balanced gradient flow across both views. Collectively, these innovations close the performance gap between the simplistic single-stream regressor and more elaborate matching-centric pipelines, while demonstrating strong robustness in ambiguous regions.

## 4 EXPERIMENTS

In this section, we describe our implementation details and evaluation protocol. Then we compare our pretrained DispViT model and the variant enhanced with external refinement module (DispViT+) to state-of-the-art methods in terms of accuracy and robustness. Then we ablate the design choices.

**Implementation details.** Unless otherwise specified, we adopt the following implementation. Our model uses a ViT-L backbone initialized with DepthAnythingV2 (DAv2) (Yang et al., 2024) weights. **Tokenizer:** The shift-embedding tokenizer shifts the right image $K = 8$ times, with each shift offset by 24 pixels, resulting in an embedding dimension of $d/8$ for each shifted view, where $d$ is the ViT-L channel dimension. **Parameterization:** Disparity is represented as a probability distribution over 128 bins uniformly discretizing the range $[0, 381]$. The loss weight $\lambda_1$ in Equation 4 is set to 0.1. **Position encoding:** We employ disparity-aware rotary position embeddings (DA-RoPE) with asymmetric frequencies—100 for the vertical direction and 1000 for the horizontal direction—to better capture the geometric priors of stereo imagery. **Refinement:** The refinement module adopts the feature extractor and refinement network of NMRF (Guan et al., 2024), while discarding its disparity proposal network and multi-hypothesis inference components to focus solely on refinement capability. **Training:** All models are trained on image crops of size $392 \times 768$. The single-stream regression model is first pretrained on a mixed dataset, consisting of FSD (Wen et al., 2025), Scene Flow (Mayer et al., 2016), TartanAir (Wang et al., 2020), CREStereo (Li et al., 2022), In-Stereo2K (Bao et al., 2020), FallingThings (Tremblay et al., 2018), Sintel (Butler et al., 2012), and Virtual KITTI 2 (Cabon et al., 2020).

**Evaluation protocol.** We evaluate across five representative datasets to assess performance under both controlled synthetic and challenging real-world conditions. For large-scale synthetic evaluation, we use the Scene Flow dataset (Mayer et al., 2016), which provides over 35,000 training pairs and 4,370 testing pairs at $540 \times 960$ resolution, spanning diverse scenarios from the FlyingThings3D, Driving, and Monkaa subsets. Real-world performance is assessed on the KITTI 2012 (Geiger et al., 2013) and KITTI 2015 (Menze & Geiger, 2015) benchmarks, including 194/195 and 200/200 training/testing pairs, respectively, with sparse and inherently **noisy** LiDAR-based ground truth from urban driving scenes. To probe cross-domain generalization, we further conduct zero-shot evaluation on the training set of Middlebury V3 (Scharstein et al., 2014), which offers high-resolution indoor scenes with dense structured-light annotations, and ETH3D (Schops et al., 2017), comprising grayscale stereo pairs of indoor and outdoor environments with challenging low-texture regions.

Table 1: **Quantitative evaluation on Scene Flow test set.** DispViT delivers competitive performance, even surpassing the ground-breaking RAFT-Stereo. A lightweight refinement (DispViT+) boosts accuracy, highlighting the robustness of DispViT as a regression prior. ♣: matching-centeric methods, ♠: hybrid methods. †: Benchmarked on RTX 3090.

| | Method | EPE ↓ | BP-1 ↓ | Time† [s] |
|---|---|---|---|---|
| ♣ | RAFT-Stereo | 0.56 | 6.63 | 0.40 |
| | DLNR | 0.48 | 5.39 | 0.33 |
| | Selective-IGEV | 0.44 | 4.98 | 0.25 |
| | NMRF | 0.45 | 4.50 | 0.10 |
| ♠ | DEFOM-Stereo | 0.42 | 5.10 | 0.63 |
| | BridgeDepth | 0.37 | 3.67 | 0.14 |
| | DispViT (Ours) | 0.53 | 5.30 | 0.092 |
| | DispViT+ (Ours) | **0.34** | **3.50** | 0.118 |

Table 2: **Ablation studies.** Upper: removal studies, where each component is individually removed from the baseline to quantify the necessity. Lower: addition studies, where components are incrementally incorporated into the baseline model to isolate their contributions.

| Model | EPE ↓ | BP-1 ↓ | Time [s] |
|---|---|---|---|
| Baseline (ViT-B) | 0.89 | 10.05 | 0.039 |
| - No DAv2 (scratch) | 1.81 | 20.13 | 0.039 |
| - No DAv2 (DINOv2) | 0.92 | 10.79 | 0.039 |
| - No probability | 1.07 | 15.56 | 0.027 |
| - No asymmetric init | 0.97 | 11.88 | 0.039 |
| - No RoPE (APE) | 0.96 | 13.34 | 0.042 |
| *addition studies* | | | |
| + shift-embedding (SE) | 0.84 | 9.22 | 0.040 |
| + DA-RoPE (DA) | 0.82 | 8.84 | 0.042 |
| + asymmetric freq (AF) | 0.76 | 8.27 | 0.042 |

Our evaluation adheres to established protocols in stereo benchmarking (Mayer et al., 2016; Menze & Geiger, 2015; Wen et al., 2025). We compute three standard metrics over all valid pixels: (1) **End-Point Error (EPE)**, the mean absolute disparity error in pixels; (2) **Bad-Pixel Rate (BP-$X$)**, the percentage of pixels whose absolute error exceeds $X$ pixels; and (3) **D1**, the official KITTI 2015 metric which measures the percentage of pixels with an absolute error greater than 3 pixels *and* exceeding 5% of the ground-truth disparity.

### 4.1 COMPARISON TO THE STATE-OF-THE-ART

The core contribution of this work is to introduce a competitive regression-centric paradigm as an alternative to the long-standing dominance of matching-centric approaches. To validate this shift, we compare DispViT with leading matching-centric methods and hybrid methods that synthesize monocular regression and stereo matching. Beyond standalone regression, we further demonstrate that complementing DispViT with a lightweight refinement network achieves state-of-the-art accuracy, while retaining the robustness and efficiency inherent to the regression-centric paradigm.

**Scene Flow.** For the Scene Flow benchmark, we follow the convention of restricting evaluation to pixels with ground-truth disparities up to 192 pixels. In this evaluation, our pretrained DispViT model is first finetuned on all 35,000 training pairs. Subsequently, the refinement network is trained from scratch with the DispViT frozen. As shown in Table 1, our single-stream regression model achieves performance comparable to leading matching-centric pipelines, *e.g.*, RAFT-Stereo (Lipson et al., 2021). Enhanced with a lightweight refinement module (∼25 ms), DispViT+ outperforms them with a notable margin. Since the refinement module is directly borrowed from NMRF (Guan et al., 2024), the substantial performance gain (**+24%**) of DispViT+ over NMRF indicates that **the improvement stems from the robustness of DispViT as a reliable regression prior rather than the refinement architecture itself**. This observation resonates with recent trends in hybrid methods like DEFOM-Stereo (Jiang et al., 2025) and Monster (Cheng et al., 2025), and more broadly establishes robust regression priors as a new cornerstone for advancing stereo disparity estimation.

**Zero-shot evaluation.** To assess the generalization of our regression-centric paradigm, we conduct zero-shot evaluation on the training sets of four real-world datasets: KITTI 2012/2015, Middlebury, and ETH3D. We compare our pretrained DispViT against strong contemporary baselines, including Selective-IGEV, NMRF, DEFOM-Stereo, and IGEV++ (Xu et al., 2024). While DispViT (ViT-L backbone) is empowered by large-scale pretraining, which differs from the training regimes of prior methods, we report full quantitative results for transparency and reference. As shown in Table 3, our approach achieves consistently competitive performance across all four benchmarks without dataset-specific fine-tuning, demonstrating strong cross-dataset generalization. Our backbone is initialized from DAv2 and further adapted to stereo disparity estimation through large-scale binocular pretraining, which we find essential for robust geometric reasoning under distribution shift. We

Table 3: **Zero-shot generalization evaluation.** [†]: Trained with extra datasets besides SceneFlow.

| Methods | KITTI-12 (D1) | KITTI-15 (D1) | Middlebury (Q) (BP-2) | ETH3D (BP-1) |
|---|---|---|---|---|
| RAFT-Stereo | 4.7 | 5.5 | 9.4 | 3.3 |
| NMRF | 4.2 | 5.1 | 7.5 | 3.8 |
| IGEV++ | 5.1 | 5.9 | 7.8 | 4.1 |
| DEFOM-Stereo | 3.8 | 5.0 | 5.7 | 2.4 |
| BridgeDepth | 3.6 | 4.5 | 4.3 | 1.3 |
| Monster | 3.6 | 4.0 | 5.1 | 2.0 |
| Monster[†] | 3.0 | 3.2 | 2.9 | 1.2 |
| DispViT | 3.9 | 4.1 | 5.5 | 4.9 |
| DispViT+ | 3.2 | 3.5 | 2.4 | 2.3 |

Table 4: **Results on ETH3D leaderboard.** Our method achieves competitive performance on the test set.

| Methods | AvgErr (px) | RMSE (px) |
|---|---|---|
| RAFT-Stereo | 0.17 | 0.42 |
| Selective-IGEV | 0.15 | 0.57 |
| IGEV++ | 0.19 | 0.74 |
| DEFOM-Stereo | 0.11 | 0.26 |
| BridgeDepth | 0.11 | 0.26 |
| Monster | 0.13 | 0.47 |
| FoundationStereo | 0.13 | 0.61 |
| DispViT+ | 0.12 | 0.25 |

anticipate that further scaling the diversity and realism of stereo pretraining data will continue to improve generalization, particularly for challenging real-world phenomena that remain underrepresented in existing synthetic corpora. Qualitative results in Figure 2 further highlight the robustness of DispViT. The model performs reliably in low-texture regions ("black holes") and on challenging surfaces such as reflections and transparency, where matching-centric methods like Selective-IGEV and NMRF often fail due to ambiguous correspondences. In contrast, the hybrid method DEFOM-Stereo, despite leveraging DAv2 estimation as initialization, does not fully retain the robustness of its monocular prior, likely due to its iterative refinement remains fundamentally matching-driven. Nonetheless, as shown in the last row of Figure 2, DispViT remains susceptible to extreme visual illusions, such as glass-induced mirror reflections, which are rare in current synthetic datasets. This limitation motivates future work on scaling stereo pretraining data or distilling the strong prior of monocular depth models to stereo with monocular disparity as affine-invariant supervision to further improve robustness.

**Kitti benchmark.** For evaluation on the KITTI benchmarks, we freeze the pretrained DispViT model and train only the refinement module from scratch using the combined KITTI 2012 and KITTI 2015 training sets. Unlike synthetic datasets with dense and accurate ground-truth disparity, KITTI annotations are sparse and often **noisy**, especially near object boundaries. Similar annotation challenges persist in other benchmarks, *e.g.*, ETH3D lacks ground truth for the non-Lambertian surfaces where robustness is most critical (last row in Figure 2). This setup presents a particular challenge for our method (DispViT+): the frozen regressor cannot adapt to dataset-specific noise, leaving the lightweight refinement to cope with the noisy labels while simultaneously correcting initialization errors from the regressor. To strengthen its capacity for KITTI noise adaptation, we employ a simple strategy, i.e., stacking the refinement network twice. As shown in Table 5, DispViT+ achieves superior or competitive performance compared to state-of-the-art methods. We emphasize that the goal of this evaluation is not to chase leader-board rankings by overfitting to dataset-specific noise, but rather to validate the effectiveness of our regression-centric paradigm in real-world scenarios.

**ETH3D benchmark.** We evaluate DispViT+ on the official ETH3D benchmark, a widely adopted testbed for stereo disparity estimation featuring challenging indoor and outdoor scenes with fine structures, textureless regions, and reflective surfaces. As shown in Table 4, our approach achieves competitive performance on this benchmark, demonstrating the effectiveness of direct disparity regression in complex real-world scenarios. Notably, this accuracy is attained with improved inference efficiency compared to state-of-the-art matching-centric pipelines (see Table 1), as our single-stream design eliminates explicit cost-volume construction and costly iterative refinement. Together, these results validate that a regression-centric formulation can produce high-fidelity disparity estimation with practical runtime efficiency, offering a streamlined yet powerful alternative to the conventional matching-driven architectures.

**Booster.** Finally, we conduct a qualitative analysis on the test set of Booster dataset (Zama Ramirez et al., 2022) to further examine the behavior of DispViT in complex indoor environments featuring highly reflective and transparent objects. Representative examples in Figure 4 highlights the robustness of DispViT under such challenging conditions, while also revealing its limitation in cases involving strong mirror-induced visual illusions.

## 4.2 ABLATION STUDY

We conduct experiments to examine the impact of each proposed design choice. Unless otherwise mentioned, all experiments use the Scene Flow dataset for training and evaluation, with a ViT-B backbone to keep the ablation study more affordable. Since certain components are critical for DispViT to reach a reasonable level of performance, we first establish a baseline configuration that achieves stable and meaningful results. This baseline is defined as a ViT-B model initialized with DepthAnythingV2 (DAv2) weights, combined with probability-based parameterization of disparity (Sec. 3.2), standard RoPE, and asymmetric initialization (Sec. 3.1). From this baseline, we perform two complementary analyses: (1) removal studies, where individual components are removed to quantify their necessity, and (2) addition studies, where components are incrementally incorporated into the baseline model to isolate their contributions.

**Removal studies.** The results of our removal studies are summarized in the upper part of Table 2. Removing pretrained weights, *i.e.*, training from scratch, causes the most dramatic degradation, with clear signs of overfitting. This underscores the necessity of large-scale pretraining not only for convergence but also as a powerful regularizer, consistent with findings in foundational vision models. We also find that probability parameterization of disparity is essential: by imposing a well-structured output space, it stabilizes training and yields a substantial performance gain (**+17%**), though at the cost of a ∼30% latency overhead. Moreover, both asymmetric initialization of the stereo tokenizer and RoPE prove indispensable, each contributing ∼10% EPE reduction by preventing early training collapse and injecting inductive biases aligned with stereo geometry, respectively. Finally, we observe that initializing from DAv2 slightly outperforms DINOv2, suggesting that geometry-aware pretraining provides a stronger prior for stereo disparity estimation.

**Addition studies.** The results of our addition studies are reported in the lower part of Table 2. We incrementally integrate three proposed designs into the baseline: shift-embedding, disparity-aware RoPE (DA-RoPE), and asymmetric RoPE frequency (assigning higher frequencies to the horizontal axis to better capture horizontal displacements).

Each component yields consistent gains, and together they contribute a cumulative **+15%** improvement. Notably, shift-embedding (SE) and DA-RoPE (DA) specifically enhance performance at large disparities, mitigating a core weakness of single-stream regression models, as illustrated in Figure 3. However, we also observe a slight trade-off with shift-embedding: while significantly improving large-disparity estimation, it marginally degrades accuracy for small disparities (< 32 pixels), likely due to reduced channel capacity per shifted view, which compromises fine-grained details necessary for small displacement estimation.

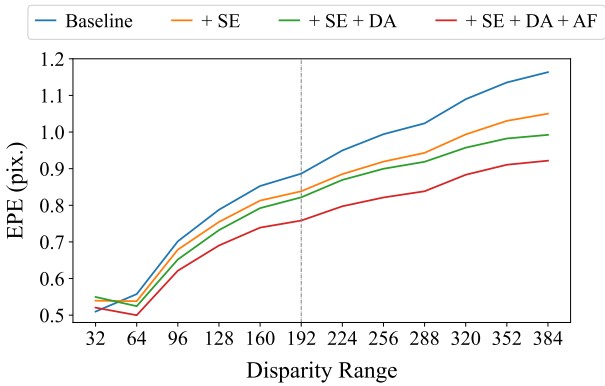

Figure 3: **End-point error (EPE) across disparity ranges.** Shift-embedding (SE) and DA-RoPE (DA) specifically improve large-disparity estimation, while asymmetric RoPE frequency (AF) yields consistent gains across all ranges by better encoding horizontal stereo geometry.

These ablation studies provide key insights into the architectural requirements for effective single-stream disparity regression. The results underscore that our lightweight designs act synergistically to meet the core demands of single-stream disparity regression, thereby solidifying the regression-centric paradigm as a powerful alternative to matching-based pipelines.

## 5 CONCLUSION AND FUTURE WORK

We presented DispViT, a regression-centric architecture for stereo disparity estimation that departs from the dominant matching-based paradigm. Through a series of lightweight yet essential architectural adaptations, a single-stream Vision Transformer directly regresses disparity from binocular inputs without explicit cost-volume construction or iterative refinement. Despite its conceptual sim-

Table 5: **Benchmark results on KITTI 2012/2015 datasets.** Metrics for KITTI 2012 are the outlier ratio (Out-$x$) for disparities errors greater than $x$ pixels in non-occluded (Noc) and all (All) regions. For KITTI 2015, results are reported using the D1 error rate across background (BG) and foreground (FG). (†): Benchmarked on GTX 3090 GPUs.

| Method | KITTI 2012 | | | | KITTI 2015 | | | | Time† |
| | Out-2 | | Out-3 | | BG | | FG | | |
| | Noc | All | Noc | All | Noc | All | Noc | All | (s) |
|---|---|---|---|---|---|---|---|---|---|
| LEAStereo | 1.90 | 2.39 | 1.13 | 1.45 | 1.29 | 1.40 | 2.65 | 2.91 | - |
| PCWNet | 1.69 | 2.18 | 1.04 | 1.37 | 1.26 | 1.37 | 2.93 | 3.16 | - |
| ACVNet | 1.83 | 2.35 | 1.13 | 1.47 | 1.26 | 1.37 | 2.84 | 3.07 | 0.2 |
| RAFT-Stereo | 1.92 | 2.42 | 1.30 | 1.66 | - | 1.58 | - | 3.05 | 0.38 |
| IGEV-Stereo | 1.71 | 2.17 | 1.12 | 1.44 | 1.27 | 1.38 | 2.62 | 2.67 | 0.18 |
| Selective-IGEV | 1.59 | 2.05 | 1.07 | 1.38 | 1.22 | 1.33 | 2.55 | 2.61 | 0.24 |
| NMRF | 1.59 | 2.07 | 1.01 | 1.35 | 1.18 | 1.28 | 2.90 | 3.13 | **0.09** |
| Mocha-Stereo | 1.64 | 2.07 | 1.06 | 1.36 | 1.24 | 1.36 | 2.42 | 2.43 | - |
| LoS | 1.69 | 2.12 | 1.10 | 1.38 | 1.29 | 1.42 | 2.66 | 2.81 | - |
| MonSter | 1.36 | 1.75 | 0.84 | 1.09 | 1.05 | 1.13 | 2.76 | 2.81 | 0.45 |
| DEFOM-Stereo | 1.43 | 1.79 | 0.94 | 1.18 | 1.25 | 1.15 | **2.23** | **2.24** | 0.61 |
| IGEV++ | 1.36 | 1.74 | 0.89 | 1.13 | 1.07 | 2.80 | 2.80 | 2.80 | 0.48 |
| BridgeDepth | 1.32 | 1.65 | 0.83 | 1.03 | 1.05 | 1.13 | 2.73 | 2.62 | **0.14** |
| DispViT+ (Ours) | **1.26** | **1.59** | **0.82** | **1.02** | **1.04** | **1.12** | 3.10 | 3.26 | 0.15 |

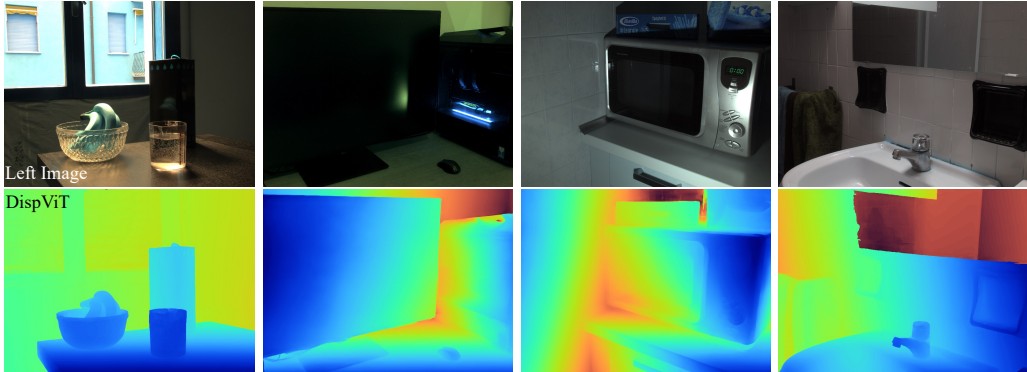

Figure 4: **Qualitative results on the challenging test set of Booster (Zama Ramirez et al., 2022).** The pretrained DispViT exhibits robustness to matching ambiguities such as transparency, low-texture, and reflections, where conventional matching methods often struggle. However, it remains susceptible to mirroring illusions (last column), a case that monocular models typically handle better.

plicity, DispViT achieves competitive benchmark performance and demonstrates strong robustness in ambiguous and low-texture regions, highlighting the promise of holistic regression for stereo reasoning. Looking forward, we plan to further enhance scalability and generalization by expanding stereo pretraining with more diverse and realistic binocular data. We are also interested in distilling strong geometric priors from large-scale monocular depth models into the stereo setting to improve robustness under distribution shift. In particular, addressing challenging mirror-induced visual illusions, currently underrepresented in existing stereo corpora, remains an important direction for strengthening real-world reliability and practical deployment.

**Acknowledgement.** This work is supported in part by the InnoHK initiative of the Innovation and Technology Commission of the Hong Kong Special Administrative Region Government via the Hong Kong Centre for Logistics Robotics, in part by the CUHK T Stone Robotics Institute, in part by and in part by the HK ITF PRP under Grant PRP/038/23FX.

The authors thank DeepSeek and GPT-5 language models for their assistance in polishing the presentation of the methodology section.

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
