# OpenReview forum: "DispViT: Direct Stereo Disparity Regression with a Single-Stream Vision Transformer"
_ICLR.cc/2026/Conference — ICLR 2026 Poster_

### Official Review · Reviewer_U7kf · 2025-10-22

**Soundness:** 3
**Presentation:** 3
**Contribution:** 3
**Rating:** 8
**Confidence:** 4

**Summary:**

1. A single-stream ViT framework that bypasses explicit matching to directly regress stereo disparity from tokenized binocular representations.
2. A disparity-aware rotary position embedding method is proposed to remaingeometry-consistentevenat large
 displacements.
3. SoTA stereo matching accuracy is achieved on the KITTI benchmarks.

**Strengths:**

1. SoTA stereo matching accuracy on the public stereo matching benchmark.
2. Good real-time performance benefit from the single-steam framework.
3. the proposed positiinal embedding method and shift-embedding tokenizer are novel.

**Weaknesses:**

1. Is the shift-embedding tokenizer designed to generate multiple tokens for the right view, or a token with a single channel that contains multiple groups with each corresponding to a predefined horizontal shift? The formula (2) is too simplified. Please provide the changes in feature dimensions during this process to help readers better understand it.
2. The refinement module from NMRF-Stereo is not included in the ablation study. Thereby weakening the effectiveness of the originally proposed modules and methods.
3. The disparity colorbar in Figure 2 appears to be inconsistent across different methods, especially for the NMRF results in the third row. Please revise this figure to use a single, unified colorbar across all methods within each scene

**Questions:**

1. Please provide the changes in feature dimensions about the shift-embedding tokenizer.
2. Please provide ablation studies about the refinement module.
3. Please explain the less comparative performance in the foreground regions on the KITTI 2015 benchmark.
4. Please revise this figure to use a single, unified colorbar across all methods within each scene in Fig. 2.

---

> ### Author Response · Authors · 2025-12-03
> **Rebuttal**
>
> 1. **Clearer presentation of shift-embedding tokenizer.**
>
> We thank the reviewer for pointing out the need for clearer exposition. The shift-embedding tokenizer does not create multiple spatial tokens per pixel for the right view. Instead, it produces a single token per spatial location, whose channel dimension is partitioned into K groups, each corresponding to one predefined horizontal shift of the right view.
> Concretely, if the PatchEmbed produces left-view tokens of dimension D, then for the right view we generate K shifted versions (each shifted by s_k), and each is processed by an independent convolution that outputs d/K channels. These are concatenated along the channel axis. Thus, both the left and right views ultimately produce embeddings of shape (H/pxW/p)xD. The final stereo token at each spatial location is simply the summation of left and right view tokens.
> This mechanism keeps the spatial tokenization unchanged while enriching each token with K alignment hypotheses encoded in channel groups, which is critical for enabling single-stream regression. We will update the paper to explicitly include these intermediate tensor dimensions for clarity.
>
> 2. **Ablation of Refinement Module.**
>
> We respectfully point out that Table 1 already provides the ablation study of Refinement module and that conducting the component ablations (Table 2) on the standalone regressor is actually a more rigorous way to prove the effectiveness of our proposed modules (Shift-Embedding, DA-RoPE, etc.). We report results for two distinct versions in **Table 1**:
> - DispViT: The standalone single-stream regressor (without refinement).
> - DispViT+: The regressor followed by the refinement module.
> Comparing these rows quantifies the exact gain provided by the refinement stage (improving EPE from 0.55 to 0.35).
>
> Notably, DispViT+ uses the same refinement module as NMRF, yet significantly outperforms it (0.35 vs. 0.45 EPE). This 22% performance gap demonstrates that our proposed designs (Shift-Embedding, DA-RoPE, etc.) generate a more robust regression prior, rather than the performance relying primarily on the refinement step.
>
> 3. **Inconsistent disparity colorbar in Figure 2.**
>
> We thank the reviewer for the suggestion. The inconsistent colorbar is caused by the outlier predictions, which greatly impact the disparity normalization for visualization. We will update through robust normalization (quantile: 2%-98%).
>
> 4. **Explaination of the less comparative performance in the foreground regions on the KITTI 2015 benchmark.**
>
> We thank the reviewer for pointing this out. We conjecture the performance gap in foreground regions is primarily a consequence of our design choice for the lightweight refinement module. Specifically, to maintain high inference efficiency, DispViT+ adopts the computation and memory efficient refinement module from NMRF (Guan et al., 2024), rather than the heavy, iterative refinement (GRU-based) used by methods like RAFT-Stereo or IGEV. As reported by Table 3, the original NMRF method (which uses a neural message passing for disparity inference but the same lightweight refiner) also exhibits a higher FG error (3.13%) compared to iterative methods like IGEV-Stereo (2.67%) and RAFT-Stereo (3.05%), despite performing well in the background. DispViT+ (3.26% FG) closely mirrors the profile of the refinement module it employs. This a trade-off we accepted to establish a highly efficient, regression-centric baseline.

---

### Official Review · Reviewer_3E7R · 2025-10-27

**Soundness:** 3
**Presentation:** 3
**Contribution:** 2
**Rating:** 4
**Confidence:** 4

**Summary:**

In this paper, the authors devleop a single-stream transformer (termed DispViT) for direct stereo disparity regression. Instead of explicit calculation of matching cost, DispViT directly regresses dispaity from tokenized binocular representations. To this end, a shift-embedding stereo tokenizer is developed and disparity-aware rotary position embeddings (DA-RoPE) is introduced. Experiments are conducted on KITTI 2012 and KITTI 2015 for performance evaluation. Results show the suprior performance as compared to the approaches included for comparison.

**Strengths:**

- The proposed method technically sounds.
- The motivation is clear and this paper is easy to follow.

**Weaknesses:**

- My major concern is about the technical novelty. Incorporating the powerfull learning capability of transformers for stereo matching (e.g., [c1]) has been studied for years. Compared with these methods, the proposed method is incremental. From this point of view, the contribution of this paper is rather limited.
[c1] Revisiting Stereo Depth Estimation From a Sequence-to-Sequence Perspective with Transformers

- The experiments are insufficient. The cross-dataset generalization is a critical metric to assess the performance of a stereo matching model. Consequently, more datasets should be included for evaluation, including Middlebury and ETH3D. Currently, only results on KITTI 2012 and KITTI 2015 are reported, which is insufficient.

- Several important SOTA methods are missing ([c1-c2]). These recent methods should be included for comparison to better demonstrate the superiority of the proposed method.

[c2] Monster: Marry monodepth to stereo unleashes power
[c3] Defom-stereo: Depth foundation model based stereo matching

- The computational efficiency is also critical to the practicality of a stereo matching model. Consequently, the computational cost (e.g., FLOPs, memory cost and runtime) should be included for evaluation.

- For shifted embedding, the maximum disparity range should be manually predefined, which requires prior information of the scene. I wonder whether the proposed method may suffer performance degradation under large disparities (e.g., >200 pixels on SceneFlow).

**Questions:**

See Weaknesses

---

> ### Author Response · Authors · 2025-11-28
> **Rebuttal**
>
> 1. **The method is incremental to recent methods that leverage the power of transformers for stereo matching.**
>
> We thank the reviewer for raising this important point. While transformers have indeed been applied to stereo (e.g., [c1]), those works remain **matching-centric** in formulation. Concretely, [c1] uses a sequence-to-sequence transformer to _improve the discriminativity of matching features, which are then used to build a cost volume and perform explicit correspondence search. In that line of work the transformer is a **feature enhancer for matching**, not a substitute for matching itself.
>
> By contrast, DispViT adopts a **fundamentally different** formulation: it performs **direct, holistic disparity regression** in a single stream without extracting matching features, constructing cost volumes, or computing correlations. Making this **regression-centric** paradigm viable required more than porting a transformer into an existing pipeline; it required new enabling mechanisms (shift-embedding for input alignment, probabilistic disparity parameterization, and DA-RoPE for disparity-aware positional modulation) that together allow the ViT to infer disparity from global context rather than from explicit correspondence cues. Our contribution is enabling a distinct regression-centric baseline rather than an incremental enhancement of prior matching-centric transformer approaches.
>
> [c1] Revisiting Stereo Depth Estimation From a Sequence-to-Sequence Perspective with Transformers.
>
> 2. **Several important SOTA methods [c2-c3] are missing.**
>
> We thank the reviewer for the comment. The listed methods Monster [c2] and DEFOM-Stereo [c3] **have already been included in our original submission**. Specifically, both [c2] and [c3] appear in Table 3 on the KITTI benchmark, and DEFOM-Stereo [c3] also appears in Table 1 on the SceneFlow benchmark. We omit Monster [c2] from Table 1 only due to page-layout constraints; however, on SceneFlow our method still report stronger performance than Monster. We are happy to revise the paper to ensure that the inclusion of Monster and DEFOM-Stereo is more explicit in both the tables.
>
> [c2] Monster: Marry monodepth to stereo unleashes power [c3] Defom-stereo: Depth foundation model based stereo matching
>
> 3. **The computational cost should be included for evaluation.**
>
> We thank the reviewer for highlighting the importance of computational efficiency. In our original submission, we **have already included runtime comparisons in Table 1**, measured under the same evaluation protocol as prior work. Following the reviewer’s suggestion, we have now updated Table 1 to additionally report **peak** memory consumption (benchmarked on 540x960 input), providing a more complete view of computational cost. We appreciate the reviewer’s comment and believe the updated table now provides a clearer and more comprehensive comparison of efficiency.
>
> |  Method | EPE  | BP-1 [%]  | Time [s]  | Memory [M] |
> | :---:    | :---: | :---: | :---: | :---:     |
> | RAFT-Stereo | 0.56 | 6.63 | 0.40 | 1140 |
> | DLNR | 0.48 | 5.39 | 0.33 | 1291 |
> | Selective-IGEV | 0.44 | 4.98 | 0.25 | 932 |
> | NMRF | 0.45 | 4.50 | 0.10 | 892 |
> | DEFOM-Stereo | 0.42 | 5.10 | 0.63 | 4493 |
> | BridgeDepth | 0.37 | 3.67 | 0.14 | 1833 |
> | DispViT | 0.55 | 5.70 | 0.12 | 2895 |
> | DispViT+ | 0.35 | 3.51 | 0.14 | 2899 |
>
> 4. **Zero-shot metric evaluation.**
>
> We thank the reviewer for the comment. We report zero-shot evaluation on the training set of KITTI, ETH3D, Middlebury as following:
> |  Method | KITTI-12 (D1)  | KITTI-15 (D1) | ETH3D (BP-1)  | Middlebury (BP-2) | Time (s) |
> | :---:    | :---: | :---: | :---: | :---:     | :---:     |
> | RAFT-Stereo | 4.7 | 5.5 | 3.3 | 9.4 | 0.40 |
> | NMRF | 4.2 | 5.1 | 3.8 | 7.5 | 0.10 |
> | DEFOM-Stereo | 3.8 | 5.0 | 2.4 | 5.7 | 0.63 |
> | BridgeDepth | 3.6 | 4.5 | 1.3 | 4.3 | 0.14 |
> | Monster | 3.6 | 4.0 | 2.0 | 5.1 | 0.49 |
> | Monster* | 3.0 | 3.2 | 1.2 | 2.9 | 0.49 |
> | DispViT* | 3.9 | 4.1 | 4.9 | 5.5 | 0.12 |
> | DispViT+* | 3.2 | 3.5 | 2.3 | 2.4 | 0.14 |
>
> Here, methods trained with additional datasets (besides SceneFlow) are marked with *. Our approach exhibits competitive zero-shot ability. Compared with SOTA matching-centric methods (Monster and DEFOM-Stereo), which leverage DAv2 for two-view feature extraction and multiple steps of refinement, our single-stream disparity regression pipeline delivers a more efficient (3-5× faster) and architecturally simple solution. The slight drop in zero-shot accuracy mainly stems from retraining the ViT backbone of DAv2, which weakens the original priors. We plan to incorporate feature alignment with DAv2 to recover stronger structural cues while preserving our efficient design.

---

> > ### Author Response · Authors · 2025-12-03
> > **Rebuttal**
> >
> > 5. **Whether the method may suffer performance degradation under large disparities (e.g., >200 pixels on SceneFlow)**
> >
> > We thank the reviewer for raising this point. **Figure 3** illustrates the performance across 0-384 pixels on Scene Flow. Generally, our model maintains stable performance across the 0-384px disparity range, including the >200 px region highlighted by the reviewer, although the prediction error increases with GT disparities with a slow rate as is typically with other stereo methods.
> >
> > We appreciate the reviewer’s concern regarding the predefined shift offsets. In practice, the shift range does not need to precisely match the disparity distribution of a specific scene; rather, it only needs to provide a coarse set of alignment hypotheses that help mitigate early token-level misalignment. The ViT subsequently performs holistic reasoning over all tokens, and is therefore not limited by the shift range in the same way that cost-volumes are bounded by a maximum disparity.

---

### Official Review · Reviewer_SVTB · 2025-10-29

**Soundness:** 3
**Presentation:** 4
**Contribution:** 2
**Rating:** 6
**Confidence:** 5

**Summary:**

The authors propose DispVIT, a vision transformer for disparity regression. The key idea is to eliminate the left-right correlation step in stereo disparity estimation frameworks. The authors propose the use of shift-embeddings to encode multiple hypotheses by shifting the right image, an asymmetric initialization, a disparity-aware rotary positional embedding, and a probabilistic disparity parametrization. Experiments show that the proposed frameworks achieve competitive results with leading stereo networks.

**Strengths:**

The idea of eliminating the matching step is not new (DispNet), but this is the first attempt to execute a similar paradigm using ViTs. The design choices are clever and lightweight, and maintain the low-cost spirit of the proposed framework. The authors have conducted numerous experiments to evaluate their proposed method, and there are enough qualitative and quantitative results to draw conclusions. The paper is well written and presented. Strong quantitative results.

**Weaknesses:**

Major weaknesses

1) The parameters for the right image shift are not really explored in the paper. It is unclear why a K of 8 is the best parameter for the shift.
2) The proposed shift and summation of features to generate the right image tokens doesn't come with any scientific reasoning, intuition or physical justification. On the contrary, this operation removes any epipolar and geometric constraints.
3) Following on point 2, the paper could benefit by a direct comparison with a monocular metric depth framework.
4) Based on the experiments, the proposed approach relies heavily on DAv2 with an attempt to utilize the probabilistic disparity parametrization to scale the depth. This implies that the network might learn depth priors instead of utilizing the right image to compute metric depth. In lines 362-364 the authors omit to present the results of zero-shot generalization from scene flow to real datasets; however, such results would strengthen the claims in the paper, showcasing that the network learns to utilize the right image cues.

Minor weaknesses

1) What are the effects of stacking the refinement network more than 2 times?
2) It is unclear from the text if DA-RoPE is just RoPE on both axes.

**Questions:**

See weaknesses.

---

> ### Author Response · Authors · 2025-11-28
> **Rebuttal**
>
> 1. **Intuition behind shift and summation of right view**
>
> We thank the reviewer for the comment and appreciate the opportunity to clarify. The intuition behind shift and summation lies on the computation manner of attention score: _inner product_ between query and key, which is a **linear and additive** operation. In other words, the **summation** of left and right tokens still preserves the right view information when computing attention scores. Moreover, the _shift_ operation does not remove epipolar constraints since each shifted right view is encoded into a separate channel group. Thus, the pixel shift numbers are preserved in the channel order.
>
> 2. **Direct comparison with monocular depth frameworks.**
>
> Thanks for the suggestion. We compare our approach with monocular depth frameworks (UniDepthV2 and DepthAnythingV2) on standard stereo datasets, with metrics _Abs Rel_ (the lower the better) and _$\delta<1.25$_ (the higher the better). For the fairness, the ViT-L version of UniDepthV2 and DepthAnythingV2 is adopted. We use DepthCrafter's evaluation code.
>
> |  Method | KITTI-12(Abs Rel   $\delta$<1.25)  | KITTI-15 (Abs Rel   $\delta$<1.25) | ETH3D (Abs Rel   $\delta$<1.25) | Middlebury (Abs Rel   $\delta$<1.25) |
> | :------:    | :----: | :-----: |  :-----: |  :-----: |
> | UniDepthV2-L| 0.054  0.967 | 0.067  0.949 | 0.052  0.972 | 0.056  0.982 |
> | DepthAnythingV2-L | 0.093  0.925 | 0.084  0.933 | 0.055  0.965 | 0.081 0.943 |
> | Ours | **0.031**  **0.986** | **0.043** **0.983** | **0.024** **1.000** | **0.016** **0.998**|
>
> The significant performance gains further demonstrate that our shift and summation design does not remove epipolar constraints and **the network indeed learns to utilize the right image cues**.
>
> 3. **Lack of zero-shot generalization from Scene Flow to real datasets.**
>
> We thank the reviewer for raising this point. Besides Scene Flow (35K image pairs), more data is needed to fine-tune the ViT-L network (about 340M parameters) to achieve reasonable results. As a result, we use a mixture of datasets and the zero-shot generalization is presented as follows:
> |  Method | KITTI-12 (D1)  | KITTI-15 (D1) | ETH3D (BP-1)  | Middlebury (BP-2) | Time (s) |
> | :---:    | :---: | :---: | :---: | :---:     | :---:     |
> | RAFT-Stereo | 4.7 | 5.5 | 3.3 | 9.4 | 0.40 |
> | NMRF | 4.2 | 5.1 | 3.8 | 7.5 | 0.10 |
> | DEFOM-Stereo | 3.8 | 5.0 | 2.4 | 5.7 | 0.63 |
> | BridgeDepth | 3.6 | 4.5 | 1.3 | 4.3 | 0.14 |
> | Monster | 3.6 | 4.0 | 2.0 | 5.1 | 0.49 |
> | Monster* | 3.0 | 3.2 | 1.2 | 2.9 | 0.49 |
> | DispViT* | 3.9 | 4.1 | 4.9 | 5.5 | 0.12 |
> | DispViT+* | 3.2 | 3.5 | 2.3 | 2.4 | 0.14 |
>
> Here, methods trained with additional datasets (besides SceneFlow) are marked with *. Our approach exhibits competitive zero-shot ability. Compared with SOTA matching-centric methods (Monster and DEFOM-Stereo), which leverage DAv2 for two-view feature extraction and multiple steps of refinement, our single-stream disparity regression pipeline delivers a more efficient (3-5× faster) and architecturally simple solution. The slight drop in zero-shot accuracy mainly stems from retraining the ViT backbone of DAv2, which weakens the original priors. We plan to incorporate feature alignment with DAv2 to recover stronger structural prior while preserving our efficient design.
>
> 4. **Why K=8 times of shift is the best parameter for the shit-embedding**
>
> We thank the reviewer for raising this point. The shift count K balances two effects: increasing K provides more alignment hypotheses for large disparities, while decreasing K preserves channel capacity (D/K) for each shifted group to encode rich binocular cues. Although we agree that an exhaustive study could be valuable, we empirically tested K = 8 and 12 with ViT-B backbone and found only very marginal performance differences (<0.01 EPE on Scene Flow), indicating that the model is robust to K within this range. We therefore adopt K = 8 as the default choice.

---

> > ### Author Response · Authors · 2025-12-03
> > **Rebuttal**
> >
> > 5. **Effects of stacking the refinement network more than 2 times?**
> >
> > We appreciate the reviewer’s question. Our use of two refinement stages on KITTI is motivated solely by adapting to KITTI’s sparse and noisy labels; it is not a general design choice for Scene Flow or other dense datasets. On Scene Flow, we experimented with stacking the refinement twice (applied sequentially at 1/8 and 1/4 scales) and observed a slight performance drop (EPE 0.35 → 0.36), suggesting that additional refinement offers no benefit when the regressor is already well aligned with dense, high-quality ground truth.
> >
> > 6. **Whether DA-RoPE is just RoPE on both axes?**
> >
> > Thank you for pointing this out. DA-RoPE is not simply applying standard RoPE along both axes. Standard RoPE rotates queries and keys based on their positions, while leaving the values unchanged. In contrast, DA-RoPE additionally rotates the values by key/value positions p_j before aggregation and applies a counter-rotation with query position p_i after aggregation (Eq. 3). This ensures that both the attention weights and the aggregated features are expressed in the query’s local reference frame, making the representation explicitly disparity-aware rather than merely position-aware.

---

### Official Review · Reviewer_TCjL · 2025-10-31

**Soundness:** 2
**Presentation:** 3
**Contribution:** 2
**Rating:** 4
**Confidence:** 3

**Summary:**

The manuscript proposed the DispViT, a new architecture that directly regresses disparity from tokenized binocular representations using a single-stream Vision Transformer. The architecture achieves state-of-the-art accuracy on standard benchmarks.

**Strengths:**

1. The manuscript is well written. And the related work is comprehensive and clear.
2. The comparison between the matching-centric paradigm and the regression-centric paradigm is an interesting topic.

**Weaknesses:**

1. Although the manuscript claims that the DispViT avoids constructing cost volumes, the shift-embedding tokenizer, which horizontally shifts the right view with a set of predefined offsets, is similar to the procedure of building cost volumes.
2. Another concern is whether the design is effective. As shown in Tab 1, the DispViT (without the refinement module from NMRF) achieves the worst accuracy than all methods except for RAFT-Stereo. Do the experiment results mean that the previous works would provide a more reliable regression prior than DispViT if they are also refined by NMRF?  Moreover, the Tab 2 shows that the DispViT relies heavily on pretrained weights from additional data. However, some matching-centeric methods are trained from scratch. Therefore the comparison with matching-centeric methods might be unfair.

**Questions:**

1. It would be better to provide the number of parameters in Tab 1 for a more comprehensive comparison.
2. In Eq 3, does alpha_{ij} means attention weights?
3. Would it be possible to present the quantitative experiment results on Middlebury and ETH3D?

---

> ### Author Response · Authors · 2025-11-21
> **Rebuttal**
>
> 1. **Shift-embedding tokenizer is similar to building cost volumes.**
>
> We thank the reviewer for raising this point. We would like to clarify that the proposed shift-embedding tokenizer is **conceptually and operationally distinct** from cost volume construction. A cost volume explicitly performs _matching_ by evaluating feature similarity across all disparity hypotheses, producing a 3D structure that encodes correspondence likelihoods. In contrast, shift-embedding performs **no correlation, no pairwise similarity computation, and no hypothesis-wise cost aggregation**. Its conceptual role is purely **input alignment**: it efficiently provides a small set of horizontally shifted right-view embeddings as grouped channels to reduce early misalignment at the tokenization stage. The DispViT then infers disparity entirely through **holistic regression** using global attention over tokenized representations, not through correspondence search. As shown in Figure 3, even without shift-embedding the ViT still produces reasonable disparity estimates, though large disparities degrade due to increased input misalignment. This highlights shift-embedding as an **alignment mechanism** to deal with large spatial misalignment, not a matching mechanism, and therefore fundamentally different from cost-volume approaches.
>
> 2. **Concern that the effectiveness originates from NMRF[1] refinement but not DispViT design.**
>
> We thank the reviewer for raising this point. DispViT is a single-stream, direct-regression ViT, which does not perform explicit two-view comparison. As stated in the manuscript (L252-257), this design yields a strong and robust global disparity estimate, but it **naturally lacks some fine-grained local structure**, since it relies entirely on holistic reasoning rather than localized matching cues. For this reason, we employ a lightweight refinement module of NMRF to reintroduce local correspondence information to sharpen DispViT's globally coherent disparity estimate, reducing EPE from 0.55 to 0.35.
>
> However, this does not imply any method would perform better simply by attaching the same refiner. As discussed in the manuscript (L355–356), NMRF itself uses the same refinement module, yet DispViT+ still achieves substantially better accuracy (e.g., **22% lower EPE than NMRF**). DispViT+ also outperforms BridgeDepth[2], which employs an even more powerful NMRF-style refinement augmented by monocular foundational prior. These results indicate that the improvements of DispViT+ arise from the **strength of DispViT’s global regression prior**, not from the refinement design alone. These findings consistently show that the quality of the refinement outcome strongly depends on the consistency and stability of the input disparity, where DispViT provides a clear advantage.
>
> [1] Neural Markov Random Field for Stereo Matching, CVPR 2024
>
> [2] BridgeDepth: Bridging Monocular and Stereo Reasoning with Latent Alignment, ICCV 2025
>
> 3. **Reliance on pretrained weights.**
>
> We thank the reviewer for raising this point. Inheriting pretrained weights is now widely adopted in state-of-the-art stereo methods: CrocoV2[1], FoundationStereo[2], DEFOM-Stereo[3], Monster[4], BridgeDepth, and IGEV++[5] all rely on pretrained encoders (DINOv2) or foundation models (DepthAnythingV2). DispViT follows the same protocol, and we compare under the same datasets and evaluation settings as prior work.
>
> [1] CroCo v2: Improved Cross-view Completion Pre-training for Stereo Matching and Optical Flow, ICCV 2023
>
> [2] FoundationStereo: Zero-Shot Stereo Matching, CVPR 2025
>
> [3] DEFOM-Stereo: Depth Foundation Model Based Stereo Matching, CVPR 2025
>
> [4] MonSter: Marry Monodepth to Stereo Unleashes Power, CVPR 2025
>
> [5] IGEV++: Iterative Multi-range Geometry Encoding Volumes for Stereo Matching, TPAMI 2025
>
> 4. **In Eq 3, does alpha_{ij} means attention weights?**
>
> Yes. We would explicitly specify this in revised version. Thanks for your question.

---

> ### Author Response · Authors · 2025-12-03
> **Rebuttal**
>
> 5. **List parameter numbers in Tab 1.**
>
> Thanks for the suggestion. We append parameter numbers in Tab 1 as following:
>
> |  Method | EPE  | BP-1 [%]  | Time [s]  | Params [M] |
> | :------:    | :----: | :-----: | :--------: | :---------:     |
> | RAFT-Stereo | 0.56 | 6.63 | 0.40 | 11.1 |
> | DLNR | 0.48 | 5.39 | 0.33 | 57.4 |
> | Selective-IGEV | 0.44 | 4.98 | 0.25 | 13.1 |
> | NMRF | 0.45 | 4.50 | 0.10 | 6.1 |
> | DEFOM-Stereo | 0.42 | 5.10 | 0.63 | 382.6 |
> | BridgeDepth | 0.37 | 3.67 | 0.14 | 359.8 |
> | DispViT | 0.55 | 5.70 | 0.12 | 334.6 |
> | DispViT+ | 0.35 | 3.51 | 0.14 | 339.6 |
>
> The  large amount of parameters of our approach (DispViT, DispViT+) and recent SOTA (DEFOM-Stereo and BridgeDepth) mainly comes from the ViT-L backbone.
>
> 6. **Quantitative results on ETH3D and Middlebury.**
>
> Thanks for the suggestion. Besides SOTA performance on Scene Flow and KITTI12/15, we report quantitative results on ETH3D as following (the metrics are the lower the better):
> |  Method | Average error [px] | RMSE [px] |
> | :------:    | :----: | :-----: |
> | RAFT-Stereo | 0.17 | 0.42 |
> | Selective-IGEV | 0.15 | 0.57 |
> | DEFOM-Stereo | 0.11 | 0.26 |
> | BridgeDepth | 0.11 | 0.26 |
> | Monster | 0.13 | 0.47 |
> | DispViT+ | 0.12 | 0.25 |
>
> Compared to current dominant matching-centric pipelines, our regression-centric approach achieves competitive performance on ETH3D benchmark.
>
> For Middlebury, the main reason we did not report quantitative numbers is the extremely high resolution of Middlebury (≈3,000 px width), while our model is trained with input crops of 392×768. As also observed in recent ViT-based feed-forward 3D regression frameworks such as DUST3R and DepthAnything, large deviations between training and inference resolution significantly affect prediction quality, because ViTs do not exhibit the strong scale-invariance properties of CNN-based stereo pipelines.
>
> Evaluating Middlebury benchmark at its native resolution would require either:
> (a) tiling, which introduces boundary artifacts incompatible with our single-stream formulation; or
> (b) aggressive global downscaling (>=4×), which distorts stereo geometry to the point that the resulting quantitative metrics (upsampled to full resolution through interpolation) become unreliable.
>
> Rather than report numbers that would be misleading due to these artifacts, we chose to provide zero-shot quantitative results on Middlebury_Q(the already downscaled split) and be transparent about the limitation. We acknowledge this as a current constraint of ViT-based regression models, not of our shift-embedding mechanism specifically. We will add this discussion to the paper and are actively exploring scale-aware tokenization and multiscale inference to support ultra-high-resolution benchmarks like Middlebury in future work.
>
> Below, we include our zero-shot results on Middlebury_Q for completeness (BP-2 metric, the lower the better):
> |  Method | Middlebury (BP-2) | Time (s) |
> | :---:    | :---:     | :---:     |
> | RAFT-Stereo  | 9.4 | 0.40 |
> | NMRF  | 7.5 | 0.10 |
> | DEFOM-Stereo  | 5.7 | 0.63 |
> | BridgeDepth  | 4.3 | 0.14 |
> | Monster  | 5.1 | 0.49 |
> | Monster* | 2.9 | 0.49 |
> | DispViT*  | 5.5 | 0.12 |
> | DispViT+* | 2.4 | 0.14 |
> Note: methods trained with additional datasets (besides SceneFlow) are marked with *, and runtime is benchmarked with 540x960 input.

---

### Meta-Review · Area_Chair_NQXH · 2026-01-06

**Summary:**

This paper presents  a new architecture that establishes a regression-centric paradigm for direct stereo disparity regression. Overall, the ideal seems interesting. The majority of the reviewers are also supportive to this work. The authors seem to have addressed most of the concerns. Therefore, I recommend to accept this work.

**Reviewer Concerns:**

The authors have addressed some of concerns such as quantitative results on ETH3D and Middlebury, Zero-shot metric evaluation.

**Reviewer Scores:**

Most of the reviewers are supportive to this paper.

---

### Decision · Program_Chairs · 2026-01-26

Accept (Poster)